# Postoperative Prevention of Urinary Tract Infections in Patients after Urogynecological Surgeries—Nonantibiotic Herbal (Canephron) versus Antibiotic Prophylaxis (Fosfomycin Trometamol): A Parallel-Group, Randomized, Noninferiority Experimental Trial

**DOI:** 10.3390/pathogens12010027

**Published:** 2022-12-23

**Authors:** Sara Wawrysiuk, Tomasz Rechberger, Agnieszka Kubik-Komar, Aleksandra Kolodynska, Kurt Naber, Pawel Miotla

**Affiliations:** 12nd Department of Gynaecology, Medical University of Lublin, ul. Jaczewskiego 8, 20-954 Lublin, Poland; 2Department of Applied Mathematics and Computer Science, University of Life Sciences in Lublin, ul. Głęboka 28, 20-950 Lublin, Poland; 3Department of Urology, Technical University of Munich, Arcisstrasse 21, 80333 Munich, Germany

**Keywords:** urinary tract infection, antibiotic resistance, urinary incontinence, pelvic organ prolapse

## Abstract

Introduction and hypothesis: Urinary tract infections (UTIs) are one of the most common complications of urogynecological surgeries. The risk of UTIs is increased by the catheterization of the bladder, intraoperative cystoscopy, and urine retention after the procedure. Due to increasing antibiotic resistance, there is a need to search for new methods of postoperative UTI prevention. Canephron is a mixture of century herbs, lovage roots, and rosemary leaves with diuretic, spasmolytic, anti-inflammatory, antibacterial, and nephroprotective properties. The aim of this study is to demonstrate the noninferiority of Canephron versus antibiotic prophylaxis with fosfomycin trometamol (FT), based on the collective results of postoperative urine culture analyses. Methods: One hundred and twenty-five female patients were randomized into two groups before undergoing urogynecological surgeries, including a control group (n = 67), which received one dose of 3 g of FT the day after the procedure, and a study group (n = 58), which received Canephron three times a day for 14 days, starting the day after the procedure. All the patients were assessed using the Acute Cystitis Symptom Score (ACSS). Results: UTIs were observed in 6.4% of the patients. There was no statistically significant difference between the use of FT and Canephron in terms of UTIs (Chi^2 N-1 = 0.8837; *p* = ns). Additional factors, such as menopausal status and the type of procedure performed, increased the risk of developing a UTI. Factors such as the body mass index (BMI) and parity had no correlation. Conclusions: Canephron is noninferior to FT in the prevention of postoperative UTIs. The use of such a phytotherapeutic drug may help to decrease antibiotic consumption, which is closely connected to the growing trend of antibiotic resistance.

## 1. Introduction

Urinary tract infections (UTIs) are one of the most common complications of urogynecological surgeries. The risk of developing a UTI is increased by the catheterization of the bladder during and after the surgery, as well as intraoperative cystoscopy and urine retention after the procedure. Approximately 7–24% of patients who undergo urogynecological surgeries develop a postoperative UTI, despite the use of prophylactic antibiotics and other preventive measures [1]. Moreover, vaginal surgeries, such as the midurethral sling (MUS) procedure, are associated with an even higher incidence of UTIs, with the risk reaching up to 34% [2].

More than 65% of UTIs in otherwise healthy patients are caused by *Escherichia coli*, followed by *Enterococcus faecalis* (12.2%), *Klebsiella pneumoniae* (4.7%), and *Proteus mirabilis* (4.2%) [3]. The most common nosocomial infection is UTI, secondary to catheterization, with an incidence rate of significant bacteriuria of more than 5% per day [4]. The National Health Safety Network (NHSN) defines catheter-associated UTI (CAUTI) as the presence of a fever, suprapubic tenderness, or costovertebral angle pain in the context of a urine culture, with bacterial counts of ≥10^5^ cfu/mL representing no more than two species of organisms (not including fungal isolates or minor pathogens) [5]. The resistance rate of *E. coli* to common antibiotics exceeds 25%, with the rates for nosocomial urinary tract infections (NAUTI) reaching 45.5% for ciprofloxacin, 48.2% for trimethoprim/sulfamethoxazole, and 50.4% for aminopenicillin [6]. This high and increasing rate of antibiotic resistance warrants a search for new methods of postoperative UTI prevention.

The commonly used method of antibiotic prophylaxis is now being replaced with nonantibiotic preparations, such as Canephron, which is a mixture of century herbs, lovage roots, and rosemary leaves with diuretic, spasmolytic, anti-inflammatory, antibacterial, and nephroprotective properties [7]. The effectiveness of Canephron in the treatment of acute lower uncomplicated UTI has already been shown in a double-blinded, randomized trial conducted by Wagenlehner et al. The study demonstrated that Canephron is noninferior to fosfomycin trometamol (FT) in terms of additional antibiotic use [8]. The effectiveness of Canephron was also proven in a large retrospective database analysis conducted by Höller et al., where it was demonstrated to be a safe symptomatic treatment for acute lower UTI [9]. Since the main role of prophylaxis is to decrease the risk of infection caused by the presence of bacteria after a procedure, Canephron’s potential to treat symptomatic UTIs also renders it a reasonable candidate as a prophylactic drug.

Another advantage of nonantibiotic prophylaxes is the protection of the microbiome. The preoperative microbiome composition of the bladder, the loss of microbiome diversity, the presence of uropathogenic bacteria, and the depletion of *Lactobacillus* species, especially *L. iners*, are associated with an increased risk of postoperative UTI [1]. Antibiotic use, including FT, results in considerable changes in the composition of the microbiome, while Canephron has been shown to preserve it [10].

The aim of this study was to demonstrate the noninferiority of Canephron versus antibiotic treatment (fosfomycin trometamol), based on the collective results of postoperative urine culture analyses and the Acute Cystitis Symptom Score (ACSS) questionnaire for the objective assessment of UTI symptoms. The acute cystitis symptom score (ACSS) is a standardized self-reporting questionnaire used for the diagnosis of acute uncomplicated cystitis based on typical symptoms, such as frequency, urgency, and dysuria [11]. It assesses the symptoms, quality of life, and changes after therapy.

The choice of fosfomycin trometamol (FT) as the antibiotic for the control group was based on its safety profile, low resistance rate, and good activity against *E. coli*, including extended-spectrum β-lactamase (ESBL)-producing strains [12]. The global resistance rate of *E. coli* strains to FT is lower than 5% among all strains and lower than 10% in the case of ESBL-producing strains [13].

## 2. Materials and Methods

The study protocol was approved by the local institutional ethical committee (26 October 2017, KE-0254/261/2017). Female patients aged 31−86 years old who were scheduled to undergo urogynecological surgeries, such as the implantation of a midurethral sling, vaginal plastic surgery, and the Manchester operation, were included in the study. The patients were hospitalized in the second gynecology department, SPSK4, in Lublin, between September 2020 and September 2021. All the participants received information on the potential risks and adverse effects of FT and Canephron. Written informed consent was obtained from the patients prior to their enrolment.

The patients underwent a urine analysis on the morning before the surgery. Prior to the surgery, each patient received 1–2 g of cefazolin (depending on the weight of the patient) or 0.5 g of metronidazole (for patients who were allergic to cefazolin) as the standard method of surgical infection prevention. The patients were randomly allocated into groups using random permuted blocks with the strata method. One hundred and twenty-five patients were randomized into two groups, including a control group (n = 67), which received one dose of 3 g of FT the day after the procedure, and a study group (n = 58), which received 5 mL of Canephron in the form of a liquid 3 times a day for 14 days, starting the day after the procedure. All the patients were catheterized during and after the surgery for no more than 24 h. Another urine analysis was conducted 14 days after the surgery, and a urine culture was performed in cases of abnormal urine analysis results or symptoms of UTI. The study was not blinded; however, a researcher who was not involved in the performance of the procedures was tasked with collecting the data in the study to reduce the risk of observer bias.

The main exclusion criteria were symptoms of UTI before the procedure (fever ≥ 38 °C, back pain, dysuria, etc.) or a positive urine analysis result (defined as the presence of bacteriuria (>10^3^ CFU/mL) and leukocyturia on a urine test based on a midstream sample), a recurrent UTI, patients with a proven UTI within the four weeks before enrolment, any antibiotic intake within the last 6 months, immunosuppression, pregnancy, and breast feeding. Any additional treatment in the study period, such as the use of herbal drugs or supplements, anti-inflammatory drugs, spasmolytics, or cranberry juice, was prohibited. The primary endpoint of the study was a positive urine analysis result 14 days after the surgery on a follow-up visit or the presence of UTI symptoms at any point during the 14 days after the surgery. The patients were advised to contact a clinic in any case of adverse effects or UTI symptoms during the study period. In cases where the patients experienced UTI symptoms, antibiotic therapy was implemented according to a urine culture, and Canephron application was terminated. The secondary endpoint included the occurrence of adverse events during the follow-up period. All the patients were assessed using the ACSS questionnaire on the day of the procedure and during the follow-up visit. Each patient received a version of the ACSS validated in the Polish language (www.acss.world; accessed on 23 December 2022).

One hundred and forty-two patients were eligible for the study. Seven patients did not attend the control visit after two weeks. Ten patients did not meet the inclusion criteria, of whom eight were excluded because of the results of a pathological urine analysis before surgery (bacteriuria, leukocyturia) and two because of prolonged catheterization (more than 24 h) after surgery (see the flowchart in Figure 1).

### Statistical Analysis

The chi-squared test is usually performed for contingency table analyses. However, it should not be used if the smallest expected number is less than 5. We used the “N-1 chi-squared test” for 2 × 2 tables in cases of expected numbers equal to at least 1, according to Campbell’s recommendations [14,15]. A noninferiority margin of 10% was applied to compare two modes of action: prevention with antibiotic treatment versus prevention with Canephron.

In order to verify the homogeneity of the study group with respect to the applied drugs, the following statistical tests were used: the t-test for age and BMI and the Mann–Whitney U test for the parity and two independent proportions of the number of post-menopausal patients.

The Wilcoxon signed-rank test was used to verify whether the values of the studied ACSS domains changed significantly after the surgery. The responses to the survey questions in both groups (Canephron versus FT) were compared using the Mann–Whitney U test.

The statistical analysis was performed using Statistica version 13 (TIBCO Software Inc. (2017), Statistica (data analysis software system) version 13. http://statistica.io, accessed on 17 November 2022), Excel, and G-Power [16].

## 3. Results

The results showed that both groups were homogenous in terms of age, BMI, parity, and menopausal status (Table 1).

UTI was observed in 6.4% of the patients during the 14 days after the procedure. The urine culture showed *E. coli* (n = 6), *Enterococcus faecalis* (n = 1), and *Streptococcus agalactiae* (n = 1) in the performed cultures. There was no statistically significant difference between the use of FT and Canephron in terms of postoperative UTI (χ^2^ = 0.88368; *p* = 0.347). The estimated differences between the groups were within the noninferiority margin of 10%. The frequency of postoperative UTI was not correlated with BMI (χ^2^ = 0.8431; *p* = 0.359) and parity (χ^2^ = 3.3931; *p*= 0.065). We identified a positive correlation between postoperative UTI and menopausal status. The patients who had undergone menopause were significantly more prone to experiencing a UTI after the surgery (χ^2^ = 4.6057; *p* = 0.032). There was also a statistically significant difference in the rate of postoperative UTI occurrence based on the type of procedure performed. The patients who underwent vaginal surgery due to pelvic prolapse (vaginal plastic surgery or the Manchester procedure) were more likely to develop postoperative UTI than patients with stress urinary incontinence (SUI) after a midurethral sling procedure (χ^2^ = 5.5254, *p* = 0.019).

The results of the ACSS on the day before the surgery (part A) and 14 (+/- 1) days after the surgery (part B) were compared for each domain and showed a statistically significant difference in both the Canephron and FT groups. In the case of the “Differential” domain score of the ACSS in the FT group, the Wilcoxon test showed that the Ho hypothesis, assuming an equal distribution of the results before and after the operation, could not be rejected. However, the *p*-value was very close to the significant level (*p* = 0.064) (Table 2).

The distribution of the sum scores of the ACSS domains was comparable between the groups at the end of the follow-up period according to the nonparametric Mann–Whitney U test. There was no statistical significance of the response distribution between the Canephron and FT groups for the “Typical” (z_adj_= 1.205, *p* = 0.23), “Differential” (z_adj_=0.04, *p* = 0.97), and “Quality of life” (z_adj_ = 0.47, *p* = 0.64) domains.

Before performing the analysis, we verified whether the number of patients was sufficient to ensure the required power of the applied statistical tests. A power equal to 0.918 was achieved for the chi-squared test, assuming a medium effect size. The power of the Wilcoxon test was higher than 0.99, while for the Mann–Whitney test, it was slightly less than 0.8 and equal to 0.77. It is worth noting that the number of observations that were initially planned in the experiment (n = 142) yielded a power of the Mann–Whitney U test equal to 0.894. The results showed that the study had sufficient power to detect statistical significance.

## 4. Discussion

Postoperative UTI is a factor of treatment failure due to improper healing. It is also associated with the development of de novo urinary urge and recurrent stress incontinence, with the risk of reoperation being required for the purpose of mesh revision/removal [17].

Canephron is noninferior to FT in the prevention of postoperative UTI. Nonantibiotic methods of perioperative prophylaxis should be considered to avoid the development of antibiotic resistance. Canephron has already been used in a real-life setting for the treatment of uncomplicated UTI and has shown a strong potential to reduce the need for outpatient use of antibiotics [8]. In a large retrospective database analysis of patients with UTI conducted by Höller et al., 2320 Canephron-receiving patients and 158,592 antibiotic-receiving patients were compared. The results showed that there were fewer sporadic recurrences of UTI after the treatment with Canephron than after antibiotic use. No statistically significant difference between the Canephron and antibiotic treatments was observed in terms of the prescription of new antibiotics within 30 days, probability of sick leave, or the occurrence of pyelonephritis [9]. Canephron was also tested as prophylactic for UTI before a urodynamic study, focusing especially on high-risk patients [18]. In our study, Canephron did not cause any major side effects. Two patients complained of nausea and stomach pain, which did not lead to the discontinuation of the treatment. Minor gastrointestinal discomfort was the most commonly reported event in both groups (Canephron: 5,1%; FT: 5,9%).

Catheterization is considered to be one of the risk factors of postoperative UTI. It is also connected with prolonged hospitalization after surgery. In the study conducted by Xie et al., 964 female patients who underwent pelvic organ prolapse (POP) surgery were compared in terms of the catheterization period. Early catheter removal (less than 2 days) was associated with a reduced incidence of UTI and an increased risk of recatheterization [19]. The patients in our study were catheterized for not more than 24 h. During our study, the catheter was always inserted using an aseptic technique, the drainage bag was positioned below the level of the bladder to prevent reflux, and the closed drainage system was maintained to prevent infection. All the patients who were eligible for the study had a postvoid residual (PVR) of less than 100 mL at discharge. Patients who required re-catheterization were excluded from the study so that this factor would not interfere with the potential risk of UTI. When a catheter is used for a period longer than 2 days, the risk of UTI doubles, with an estimated 5% increase in the bacteriuria for each day of catheterization. Thus, leaving catheters applied for the shortest time possible after surgery can lower the incidence of postoperative UTI [20]. The most common pathogen isolated from urine during catheterization is still *E. coli*; however, it has recently shown an increased rate of resistance [21]. UTI is usually connected to biofilm formation on the surface of the catheter due to the susceptibility of catheter material to microbial adhesion [22]. Appropriate antibiotic prophylaxis can decrease the occurrence of postoperative UTI compared to placebos [23,24]. In the study conducted by van der Wall et al., in which 184 catheterized patients were randomly allocated to groups receiving either a placebo or ciprofloxacin, the patients who received ciprofloxacin had significantly less CAUTI than the patients in the placebo group. The prevalence of pyuria in the placebo group reached 42% [25].

Menopausal status had a strong statistical correlation with postoperative UTI. Due to a reduction in estrogen levels after menopause, changes tend to occur in the urogenital epithelium, such as decreased glycogen production, *Lactobacillus* depletion, decreased lactic acid production, and pH elevation [26]. This environment is more vulnerable to uropathogens and more prone to infection, especially after surgery, which causes additional tissue damage. Moreover, in postmenopausal women, the resistance of all bacteria that cause UTIs to commonly used antimicrobials has been shown to be significantly higher than that in premenopausal women [3]. In a retrospective study which reviewed 310 patients who underwent urogynecological surgeries, older postmenopausal women with prolonged voiding dysfunction who had received a greater number of concomitant pelvic floor surgeries had a higher risk of postoperative UTI [27].

Obesity is a significant risk factor for UTI. Obese patients are 2,5 times more likely to suffer from different kinds of UTIs, while the incidence is correlated with an increased BMI [28]. However, in a retrospective cohort study investigating the risk factors and prevalence of UTI one year after incontinence surgery, 178 patients were examined, and no correlation between postoperative UTI and the BMI was identified [29]. These findings corresponded with the results of our study, where the BMI did not significantly correlate with UTI occurrence.

UTI is the most common complication of POP surgery. In the study conducted by Bretschneider et al., where a total of 9546 patients underwent colpopexies due to POP, urinary tract infections occurred in 5.1% of the patients, and no significant differences were identified between the complication rates of the different approaches to vaginal colpopexy [30]. Sutkin et al. calculated a cumulative risk of symptomatic postoperative UTI for both prolapse and incontinence surgery based on 389 cases, which reached 9.0% [31]. This study showed that UTIs were more common in patients undergoing POP surgery than UI surgery, which corresponded with our results.

The ACSS questionnaire results indicated a decrease in the severity of lower UTI symptoms over time and an increase in the quality of life (QoL) of patients treated with Canephron and FT. In this case, a decrease in lower UTI symptoms 14 days postsurgery should also be understood in terms of the similarities between some lower urinary tract symptoms (LUTSs) of UTI, SUI, and POP. Some patients presented with pain, frequency, urgency, or incomplete bladder emptying due to a primary disease, which was SUI or POP, and not due to UTI. Additionally, the increase in the QoL could be explained by the alleviation of SUI and POP symptoms after the surgery, which led to less discomfort, no disruptions in everyday activity, and less interference with the patient’s social life. However, there was no statistically significant difference in influence on the UTI symptoms and QoL between the groups based on the ACSS.

The strengths of our study included the design of the study, based on randomization, and the consecutive selection of patients, which reduced the selection bias. The urine samples were examined in the same laboratory and were taken immediately before the analysis. All the operators used the same techniques when performing the surgeries. Additionally, the POP surgeries were limited to the vaginal approach alone, and the UTI surgeries were limited to the transobturator procedure. Moreover, the patients were instructed to call our facility if any UTI symptoms occurred before the control visit; thus, they did not receive any antimicrobial drugs from other doctors. However, our trial did have some limitations, including the short observation period and limited number of participants, especially in the case of those with a UTI. Given that UTIs are a frequent postoperative complication of both SUI and POP surgeries, our study could have focused more on identifying a high-risk group of patients, so that the strategies of prevention could be incorporated, especially in the case of this group. Since all the patients received a single preinterventional antibiotic dose, the individual, additional contribution of the postinterventional nonantibiotic, as well as that of antibiotic prophylaxis could not be determined. Nevertheless, in both groups, the same satisfactory results were obtained. Moreover, we must acknowledge the fact that, for some patients, taking the medication for a prolonged time (in this case, for 14 days) can be problematic in terms of adherence to the treatment regimen. We did not observe noncompliance in our study, even considering the fact that the use of one dose of medication appeared to be more straightforward. We believe that this level of compliance was obtained because the patients were well-informed about the purposes of this research.

## 5. Conclusions

Perioperative prophylaxis with Canephron should be considered as an alternative to classic antibiotic use. Canephron is a safe and effective method of UTI prevention. The use of such a phytotherapeutic drug may help to decrease antibiotic consumption, which is closely connected to the growing trend of antibiotic resistance. Together with the results of previous studies [8,9,10], this research may encourage the use of antibiotic alternatives as new methods of UTI prevention. However, more prospective studies conducted on larger groups of participants are required to show that Canephron and other phytotherapeutic drugs are safe and effective.

## Figures and Tables

**Figure 1 pathogens-12-00027-f001:**
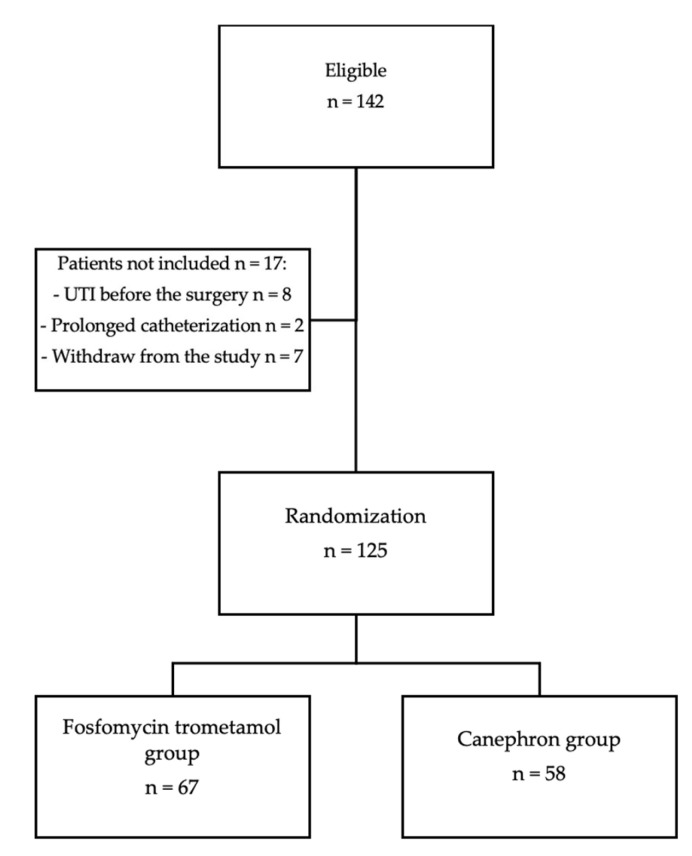
Selection of study patients.

**Table 1 pathogens-12-00027-t001:** Patients’ demographic data and *p* values.

Variable	Fosfomycin Trometamol (n = 67)	Canephron (n = 58)	*p*
Age (years), M ± SD	57.6 ± 12.5	57.7 ± 13.0	*p* = 0.989
BMI (kg/m^2^), M ± SD	28.8 ± 5.8	28.4 ± 4.8	*p* = 0.616
Parity, M ± SD	2.1 ± 0.8	2.0 ± 0.7	*p* = 0.716
Menopause, number (%)	47 (70%)	38 (66%)	*p* = 0.332

BMI—body mass index.

**Table 2 pathogens-12-00027-t002:** Mean scores in ACSS domains during first and follow-up visit with Wilcoxon signed-rank test values.

Total Scores in ACSS Domains	Fosfomycin Trometamol						Canephron					
	Day 1		Day 14		z Value	*p* Value	Day 1		Day 14		z Value	*p* Value
	Mean	SD	Mean	SD			Mean	SD	Mean	SD		
“Typical”	3.21	3.58	1.45	2.88	5.20	<0.001	2.72	2.81	0.62	1.17	4.34	<0.001
“Differential”	0.63	1.51	0.28	0.77	1.85	0.064	1.07	1.69	0.24	0.78	3.12	0.002
“Quality of life”	2.13	2.30	0.93	1.69	3.45	0.001	1.26	1.51	0.64	1.18	2.94	0.003

ACSS—acute cystitis symptom score; SD—standard deviation.

## Data Availability

The data presented in this study are available on request from the corresponding author. The data are not publicly available due to privacy restrictions.

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
