# Peer review of "Postoperative Prevention of Urinary Tract Infections in Patients after Urogynecological Surgeries—Nonantibiotic Herbal (Canephron) versus Antibiotic Prophylaxis (Fosfomycin Trometamol): A Parallel-Group, Randomized, Noninferiority Experimental Trial"

_pathogens, 2022, doi:10.3390/pathogens12010027_

Round 1
Reviewer 1 Report
Dear authors,
I have read with interest your manuscript and I have some suggestions:
1. Please move the information about statistical sample and figure 1. Selection of study patients into Materials and Methodspart from Resultspart.
2. Please specify into Materials and Methodspart the statistical tool used to obtain the results.
3. I suggest to exclude the sentences with description of statistical methods: e.g. lines 125-128,
4. Also, I suggest tomention in Conclusions part the utility of the study in future research.
5. You are mentioning that you found significant differences in UTI occurrence depending on the type of surgery, please include it in table 1.
6. I suggest to replace the “ns” results for p with values in table 1
7. Please write before period the reference numbers in the text body.
8. I suggest to move in formation describing ACSS methods and utility (line 247-250)in the introduction part
Reviewer 2 Report
Dear Authors,
After reading the manuscript entitled " Prevention of urinary tract infections in patients after urogynecological surgeries – non-antibiotic herbal (Canephron N) versus antibiotic prophylaxis (Fosfomycin Trometamol): A parallel-group, randomized, non-inferiority, experimental trial", I do not think it is suitable for printing in its present form. The manuscript after the corrections can be sent for review again.
Reviewer's comments:
lines 13, 37, 45 is: …UTI…should be: …UTIs.…
line 40 is: … urogynecologycal…, should be: … urogynecological…
lines 45-47 - Bacteria names should be in italics. Please revise this throughout the manuscript.
line 71 is: …presence……depletion…, should be: …the presence……the depletion….
line 80 - The abbreviation ESBL should be explained.
line 80 is: …ESBL producing…, should be: …ESBL-producing…
line 107 is: …Main…, should be: …The main…
line 115 is: …(>10 3 CFU/mL)…, should be: …(>103 CFU/mL)…
lines 145-147 is:… Urine culture showed Escherichia coli in 6 cultures, Enterococcus faecalis, and Streptoccocus agalactie in the remaining cultures…” – This sentence is unclear. In how many cultures were Enterococcus faecalis and Streptoccocus agalactie found? Were these two bacteria together?
line 147 is: …agalactie…should be: …agalactiae.…
lines 164-165 is: … a Wilcoxon test showed that Ho hypothesis could not be rejected… - Please, provide the content of the hypothesis H0.
lines 166-168 is: … Mean sum scores of the ACSS domains were comparable between the groups at the end of the follow-up period, there was no statistical significance between Canephron and FT group…– This sentence is unclear. Look at Total “Typical” score in FT group is 1.45 (the mean value), in Canephron group is 0.62 - this is a significant difference.
line 169-170 is: … where total “z” scores for the typical, differential and “quality of life” 169 domains were respectively 1.205, 0.042 and 0.470 (Table 2.) - It is incomprehensible. These values are not in Table 2. Explain it.
lines 172-176 – The text includes the values obtained in statistical tests. Write what conclusions can be drawn from these values. This is to be clear to the reader who is not very knowledgeable about statistics.
line 183 is: … development…, should be: …the development…
line 185 is: … for treatment…, should be: … for the treatment…
line 214 is: … due to susceptibility…, should be: … due to the susceptibility…
line 218 is: … ciprofloxacine…, should be: … ciprofloxacin…
line 251 is: … QoL…” – explain this abbreviation
line 254 is: … SUI…” – explain this abbreviation
line 267 is: … drug…, should be: … drugs…
Reviewer 3 Report
Dear Authors, thank you for giving me the opportunity to review your manuscript. The general impression made by this publication - it is “raw”, immature and perfunctory. It should be revised substantially. Please find some comments below:
1. The title is confusing as strictly speaking all patients had pre-interventional antimicrobial prophylaxis (even with Canephron B)
2. Please, provide sufficient background by defining current recommendations on antimicrobial perioperative prophylaxis in urogynecological surgeries, namely midurethral sling implantation, vaginal plastic surgery, Manchester operation
3. Please, reference the rationale for both pre-interventional and post-interventional antimicrobial prophylaxis. Is there an evidence of post-interventional antibiotics utility at all?
4. Lines 47-55: It is not clear in text, whether you show the resistance rates of uropathogens causing CAUTI or NAUTI. Please, emphasize the difference in resistance rates and treatment strategies of CAUTI and NAUTI, providing latest/more recent references
5. Put the definition of the “positive urine analysis” when it is firstly mentioned in the text (Line 108)
6. Please, explain the abbreviations, when firstly mentioned in the text (POP, UI, QoL etc.)
7. Lines 161-170: Should be in Materials and Methods, not in the Results section
8. Please, put the references appropriately (not in the beginning of sentences)
9. Microorganisms should be italicized
- Please check if the references are described as Instructions for Authors
11. Antibiotics should be from the small letter (e.g. cefazolin, metronidazole, fosfomycin trometamol). There are several words that should also be written with the small letter (cranberry, clinic etc.)
12. Add the full stops (e.g. Line 193), delete excessive spaces (e.g. Line 234) or add a space where is needed (e.g. Lines 141, 207) throughout the text.
13. Please, unify “Canephron N” spelling throughout the text.
14. Lines 91, 109: Enrollment
15. Table 1: delete “p=” in the menopause line
16. Figure 1: Catheterization
17. Lines 145-147: Put the number in brackets after each pathogen, e.g. E. coli (n=6), E. faecalis (n=…
18. Line 147: agalactiae
19. Line 148: Canephron N
20. Line 153: Patients in postmenopause
21. Lines 218, 219: ciprofloxacin
22. Lines 49, 213: change “connected” to “related”
Round 2
Reviewer 2 Report
Dear Authors,
The manuscript has been revised, but minor corrections still need to be made:
line 90 is: ...E. coli... - italics
lines 179, 251 is: ...Escherichia coli... should be:... E. coli...
lines 382, 385 is ...Escherichia coli... - italics
Author Response
Thank you very much, changes have been made.
Reviewer 3 Report
The manuscript may be accepted for publication
Author Response
Thank you very much.